# Allyl-, Butyl- and Phenylethyl-Isothiocyanate Modulate Akt–mTOR and Cyclin–CDK Signaling in Gemcitabine- and Cisplatin-Resistant Bladder Cancer Cell Lines

**DOI:** 10.3390/ijms231910996

**Published:** 2022-09-20

**Authors:** Jochen Rutz, Sebastian Maxeiner, Timothy Grein, Marlon Sonnenburg, Salma El Khadir, Nino Makhatelashvili, Johanna Mann, Hui Xie, Jindrich Cinatl, Anita Thomas, Felix K.-H. Chun, Axel Haferkamp, Roman A. Blaheta, Igor Tsaur

**Affiliations:** 1Department of Urology and Pediatric Urology, University Medical Center Mainz, 55131 Mainz, Germany; 2Department of Urology, Goethe-University, 60590 Frankfurt am Main, Germany; 3Institute of Medical Virology, Goethe-University, 60596 Frankfurt am Main, Germany

**Keywords:** bladder cancer, isothiocyanates, Akt–mTOR signaling, CDK–cyclin axis, drug resistance

## Abstract

Combined cisplatin–gemcitabine treatment causes rapid resistance development in patients with advanced urothelial carcinoma. The present study investigated the potential of the natural isothiocyanates (ITCs) allyl-isothiocyanate (AITC), butyl-isothiocyanate (BITC), and phenylethyl-isothiocyanate (PEITC) to suppress growth and proliferation of gemcitabine- and cisplatin-resistant bladder cancer cells lines. Sensitive and gemcitabine- and cisplatin-resistant RT112, T24, and TCCSUP cells were treated with the ITCs, and tumor cell growth, proliferation, and clone formation were evaluated. Apoptosis induction and cell cycle progression were investigated as well. The molecular mode of action was investigated by evaluating cell cycle-regulating proteins (cyclin-dependent kinases (CDKs) and cyclins A and B) and the mechanistic target of the rapamycin (mTOR)-AKT signaling pathway. The ITCs significantly inhibited growth, proliferation and clone formation of all tumor cell lines (sensitive and resistant). Cells were arrested in the G2/M phase, independent of the type of resistance. Alterations of both the CDK–cyclin axis and the Akt–mTOR signaling pathway were observed in AITC-treated T24 cells with minor effects on apoptosis induction. In contrast, AITC de-activated Akt–mTOR signaling and induced apoptosis in RT112 cells, with only minor effects on CDK expression. It is concluded that AITC, BITC, and PEITC exert tumor-suppressive properties on cisplatin- and gemcitabine-resistant bladder cancer cells, whereby the molecular action may differ among the cell lines. The integration of these ITCs into the gemcitabine-/cisplatin-based treatment regimen might optimize bladder cancer therapy.

## 1. Introduction

Urothelial carcinoma is the most common tumor of the urinary tract worldwide, with over 570,000 new cases and more than 210,000 deaths in 2020 [1]. The standard care for advanced and/or metastatic urothelial carcinoma is based on the cytotoxic protocols MVAC (methotrexate, vinblastine, doxorubicin, and cisplatin) or GC (gemcitabine and cisplatin) [2]. Unfortunately, despite a high initial response rate, resistance rapidly develops by reactivation of intracellular growth signaling pathways. The median survival rate of patients treated with cisplatin-based chemotherapy is limited to 14 months [3]. Meanwhile, immune checkpoint inhibitors (ICIs) have been established targeting programmed cell death-1 (PD-1) or the programmed cell death-ligand 1 (PD-L1). The European Medicines Agency (EMA) and the Food and Drug Administration (FDA) have approved the use of ICIs for second-line treatment after failure of platinum-based chemotherapy (atezolizumab, pembrolizumab, and nivolumab) or in a first-line setting for cisplatin-ineligible patients with PD-L1 positivity (atezolizumab, pembrolizumab) [4]. Still, the clinical results are not as successful as has been expected. Of note, utilization of pembrolizumab or atezolizumab in the second-line treatment was associated with an overall response rate of 15–20% and progression-free survival of slightly more than 2 months in the landmark IMvigor210 and Keynote 045 trials (PMID: 33421822) [4,5].

Dissatisfaction with the conventional tumor treatment along with severe side effects under MVAC/GC or ICIs has driven many cancer patients to seek complementary or alternative medicine (CAM). Worldwide, approximately 50% of cancer patients are reported to use CAM, whereby variations should be considered, depending on the tumor entity, patient socio-demographics and country [6]. Of several options, intake of plant extracts or plant-derived compounds is mostly favored with a percentage application rate of nearly 50% [7,8]. Indeed, clinical studies have provided evidence that the integration of plant-derived natural drugs into the conventional anti-tumor protocol may improve the efficacy of the systemic bladder cancer treatment [9,10].

The present study deals with the impact of the natural isothiocyanates (ITCs) allyl-isothiocyanate (AITC), benzyl-isothiocyanate (BITC), and phenethyl-isothiocyanate (PEITC) on the growth and proliferation properties of cisplatin- and gemcitabine-resistant bladder cancer cell lines. ITCs are abundant in cruciferous vegetables such as broccoli, gardencress, watercress and mustard and derived from sulfur-containing glucosinolates after enzymatic hydrolysis (Figure 1) [11]. BITC is predominantly found in nasturtium, whereas AITC and PEITC are typical ITCs found in horseradish root. Evidence has already been provided that AITC and BITC suppress bladder cancer growth and enhance apoptosis in vitro. Both compounds reduced the size of bladder tumors and prevented oncogenesis in a rat model as well. PEITC is also assumed to play critical roles in preventing the initiation step of carcinogenesis and inhibiting tumor progression. However, the detailed molecular mechanisms have not been clearly defined [12]. There are also no publications available dealing with the influence of the ITCs on cisplatin- and gemcitabine-resistant bladder cancer.

## 2. Results

### 2.1. Resistance Induction

Chronic treatment of T24, RT112, and TCCSUP cells with increasing concentrations of cisplatin or gemcitabine induced resistance development. The response to drug treatment is shown in Figure 2 (sensitive versus resistant cells). A dosage of 1.25 ng/mL (RT112, TCCSUP) or 2.5 ng/mL (T24) gemcitabine, respectively, was sufficient to diminish growth of the gemcitabine-sensitive cells. In contrast, higher dosages were necessary to evoke growth suppression, which were >20 ng/mL gemcitabine (20 ng/mL gemcitabine = 0.0758 µM) in the RT112 and TCCSUP model, and 40 ng/mL in the T24 cell line (Figure 2A). Cisplatin at 0.125 µg/mL (RT112, TCCSUP) or at 0.5 µg/mL (T24) already significantly reduced tumor growth in the sensitive setting (Figure 2B). Higher concentrations were applied to suppress growth of the resistant cell lines (RT112, TCCSUP: 1 µg/mL; T24: 2 µg/mL; 1 µg/mL cisplatin = 3.332 µM). Application of the trypan blue assay showed that drug treatment induced no toxic effects.

### 2.2. BITC, AITC, and PEITC Block Growth of Drug-Resistant Tumor Cells

All ITCs applied blocked growth of both the drug-sensitive (parental cells) and the drug-resistant bladder cancer cell lines (Figure 3A–C). However, the effect depended on the cell line used and the resistance status. Concerning BITC (Figure 3A), cisplatin-resistant RT112 and T24 cells responded stronger to drug treatment than the sensitive counterparts did. Particularly 12.5 µM BITC blocked growth of the cisplatin-resistant RT112 and T24 cells completely, whereas a still moderate increase in the cisplatin-sensitive RT112 and T24 cells was observed in presence of 12.5 µM BITC. Differences between the response of sensitive versus resistant T24 and TCCSUP cells were also noted with respect to AITC treatment, although higher concentrations of AITC, compared to BITC and PEITC, were necessary to induce a significant down-modulation of the tumor cell number (Figure 3B). Comparison between the three cell lines documented T24 to be the most sensitive one. A dosage of 70 µM AITC led to a complete stop of T24 growth, whereas concentrations up to 140 µM have been added to RT112 and TCCSUP cultures. PEITC also acted as a growth suppressor. Remarkably, gemcitabine-resistant TCCSUP very strongly responded to the drug treatment, and no cells were counted in presence of 12.5 µM PEITC (Figure 3C). Zero growth was also noted when cisplatin-resistant RT112 were treated with 12.5 µM PEITC. All further experiments were then performed with 7.5 µM BITC, 20 µM AITC (if not otherwise indicated), and 7.5 µM PEITC.

### 2.3. Apoptosis Induction by BITC, AITC, and PEITC

Apoptosis was evaluated in parental and drug-resistant RT112, T24, and TCCSUP cells, following a 24 and 72 h drug incubation with BITC (7.5 µM), AITC (20 and 40 µM) and PEITC (7.5 µM) (Figure 4). All cell lines responded similarly to BITC, AITC, and PEITC, whereby apoptotic events were more pronounced after 72 h, compared to the events observed after 24 h (excepting gemcitabine-resistant T24 cells treated with AITC, see below). No differences were seen between 20 and 40 µM AITC. Data are depicted representatively for RT112 (Figure 4A) and T24 cells (Figure 4B), exposed to the compounds for 72 h. BITC induced early apoptosis (particularly RT112); however, strongest effects were seen on late apoptosis. Necrosis was induced as well after 72 h (particularly T24). AITC exposure evoked prominent elevation of tumor cells undergoing late apoptosis excepting gemcitabine-resistant T24 cells, where the number of apoptotic cells were below the controls. However, following a 24 h AITC incubation, gemcitabine-resistant T24 cells undergoing early apoptosis increased to 24% (mean), and those undergoing late apoptosis were enhanced to 25% (mean). PEITC elevated apoptosis (late > early apoptosis), with only a very slight influence on tumor cell necrosis.

### 2.4. Suppression of Clonogenic Tumor Growth

To investigate the potential of BITC, AITC and PEITC to stop colony formation, the clonogenic growth assay was performed (parental and drug-resistant cells). Morphologic analysis demonstrated a compact and dense clone structure of untreated RT112 and T24 cells (shown for the sensitive cell lines). The disintegration of the tumor cell clones became obvious in presence of the compounds applied at 7.5 µM (BITC and PEITC) or 20 µM (AITC) (Figure 5).

Figure 6 depicts the clone number in treated versus non-treated RT112 (Figure 6A), T24 (Figure 6B), and TCCSUP cells (Figure 6C) exposed to 7.5 µM BITC or PEITC, or 20 and 40 µM AITC. Independent on the ITC and the cell line used, and independent on the resistance status, drug treatment was associated with a significant loss of the clone number (Figure 6A–C).

### 2.5. Cell Cycle Arrest by BITC, AITC, and PEITC

Distinct cell cycle alterations were evoked by BITC, AITC and PEITC on both the parental and drug-resistant cells. Overall, there was no difference between 24 and 72 h treated T24 cells. However, RT112 cells responded well to BITC and PEITC after 72 h, whereas strongest response to AITC was noted after 24 h. Therefore, Figure 7A presents both 24 and 72 h data sets for T24 cells and RT112 results recorded after 72 h (BITC and PEITC) or after 24 h (AITC), respectively. The proportion of G2/M-phase RT112 cells significantly increased, whereas the number of G0/G1-phase cells decreased in presence of the compounds. The number of S-phase RT112 cells increased as well when exposed to BITC or PEITC, but was diminished following AITC application. Similar effects were evoked by the compounds on T24 cells (Figure 7B), excepting the cisplatin-resistant sub-line, where AITC induced only a moderate (24 h incubation) or no alteration (72 h incubation) on the percentage of G2/M-phase cells. There was also no influence of BITC on parental G0/G1-phase T24 cells after 72 h.

### 2.6. Alteration of Cell Cycle Protein Expression

Subsequent studies concentrated on cell cycle regulating protein expression in parental, cisplatin- and gemcitabine-resistant RT112 and T24 cells (Figure 8 and Appendix A). Protein alterations were detected after 24 or 72 h in T24 cells. Concerning RT112, maximum effects were seen after 72 h in presence of BITC and PEITC, whereas strongest modulation was evoked by AITC after 24 h. The overall picture is inhomogenous. Concerning T24 cells, pAkt was strongly reduced by AITC and PEITC (no signal detected in presence of BITC). Total Akt was also reduced in parental cells when exposed to PEITC. pRictor was considerably up-regulated, whereas pRaptor was strongly diminished in AITC-treated cells. CDK1 and CDK2 were reduced by all compounds in T24 cells. Protein reduction was also true for pCDK1 and cyclin A in BITC-, AITC-, and PEITC-treated cells. With respect to RT112, pAkt was diminished by all compounds, excepting AITC which caused an up-regulation of pAkt in parental cells. Interestingly, pRictor was suppressed in the parental cells by BITC, AITC, and PEITC, whereas this protein was elevated in the resistant sub-lines (excepting gemcitabine-resistant RT112 treated with AITC). pRaptor decreased in the parental and gemcitabine-resistant RT112 independent on the compound used but strongly increased in the cisplatin-resistant cells under PEITC. CDK2 was suppressed in the parental and resistant RT112 cells when treated with BITC or PEITC, and diminished in the gemcitabine-resistant subline by AITC. Slight alterations of cyclins A and B were observed which depended on the drug applied and the resistance status. Figure 9 provides the relevant pixel density data.

## 3. Discussion

In vivo and in vitro data have already demonstrated that ITCs may suppress oncogenesis and tumor progression [12]. There is also epidemiological evidence that ITCs may diminish the risk of bladder cancer recurrence [13]. However, information about AITC, BITC and PEITC is sparse, and no reports are available dealing with the influence of these compounds on drug-resistant bladder cancer. Evidence is presented here that AITC, BITC, and PEITC block growth and proliferation of drug-sensitive as well as of cisplatin- and gemcitabine-resistant bladder cancer cell lines. Based on dose–response-analyses (2.5–12.5 µM BITC or PEITC, 20–140 µM AITC), working concentration was set to 7.5 µM BITC or PEITC, and to 20–40 µM AITC. Similar dosing schedules have been used by others with slight modifications depending of the cell line used. Lin et al. pointed to a significant loss of the number of RT4, 5637, HT1376, and HT1197 bladder cancer cells when treated with BITC at a concentration of 20 µM [14].

A dosage of 15–30 µM BITC or PEITC was necessary to induce damage of UMUC3 bladder cancer cells [15], whereas T24 cell proliferation was suppressed already at 2–5 µM PEITC [16]. AITC at concentrations of >60 µM has been noted to block growth and proliferation of RT4 and T24 cells [17,18]. UMUC3 cells, however, responded to a lower dosage of 15 μM AITC in terms of cell growth inhibition [19].

We assume that AITC, BITC and PEITC are all characterized by tumor-suppressive properties, whereby the tumor cells’ sensitivity may depend on the tumor cell subline. From a clinical viewpoint, bladder cancer patients may differentially benefit from ITC consumption. Independent on this, BITC and PEITC were superior to AITC in the present investigation, since growth-blocking effects had been evoked at low concentrations, whereas AITC had to be applied at higher dosages. Interestingly, comparative analysis of AITC, BITC, and PEITC on apoptosis induction revealed the strongest potency of BITC and PEITC [15].

The authors of this study assumed that the lipophilicity represents at least one parameter defining the activity of the drugs. In this context, BITC and PEITC are rather hydrophobic, whereas AITC is more water-soluble. This is important as a high lipid solubility (depending on the alkyl carbon chain of the ITCs) correlates with the transmembranous transport of the molecule into the tumor cell.

Quite importantly, AITC, BITC, and PEITC acted not only on drug-sensitive cells but also on cisplatin- and gemcitabine-resistant sublines. The response of the resistant tumor cell lines was similar to the one of the sensitive cells, highlighting the therapeutic potential of these ITCs as integrative components in a GC-based regimen. The assumption is still hypothetical and, certainly, further studies are required to verify it. However, 5–10 µM BITC or 20–40 µM AITC have already been documented to induce apoptosis and to inhibit proliferation of human cisplatin-resistant oral cancer cells [20,21], and PEITC has been proven to reverse cisplatin-resistance in cervical cancer cells at low concentrations of 1–2 µM [22].

Apoptosis induction might be one mechanism how the ITCs have reduced tumor cell growth. In good accordance, BITC elevated cytotoxicity and apoptosis of a panel of bladder cancer cell lines [14]. BITC, AITC, and PEITC all increased apoptosis and necrosis rates in RT4 and T24 cells [15,18]. PEITC has also been demonstrated to induce apoptosis in adriamycin-resistant bladder cancer cells [16], underlining the relevance of ITCs as a therapeutic strategy to fight drug resistance in bladder cancer. The apoptotic machinery has not been further evaluated in the present investigation. Based on Sávio and coworkers, the mechanism of action of BITC, AITC, and PEITC on apoptotic pathways may differ among several bladder cancer cell lines [17]. Ongoing experiments should, therefore, deal with the fine-tuned action of these ITCs on the apoptotic cascade.

Aside from apoptosis induction, ITCs caused a prominent accumulation of RT112 and T24 cells in G2/M, accompanied by a loss of G0/G1-phase cells. The effect was similar in all cell lines and the respective resistant sublines, opening the question whether this mechanism might be a general one. Indeed, G2/M-phase arrest in presence of AITC has been observed in T24 [18] and UMUC3 cell cultures by others [19]. G2/M-phase arrest has also found in cell cultures from further tumor entities treated with BITC, AITC, or PEITC [23,24,25]. Presumably, G2/M-phase cell cycle arrest observed here might be mainly responsible for the loss of bladder cancer cell growth and proliferation resulting from ITC treatment. However, BITC has also been reported to induce G1 cell cycle arrest in the Transgenic Adenocarcinoma Mouse Prostate (TRAMP) model [26] and glioblastoma multiforme cells [27].

Obviously, the present results do not allow to exclude additional functions of ITCs on cell cycle progression. In fact, the Western blot data document different alterations of cell cycle regulating proteins in RT112 and T24 cells when exposed to the ITCs. Furthermore, response depended also on the particular ITC used. Overall, CDKs and cyclins were considerably suppressed in T24 cells (being aware of few exceptions), whereas only CDKs (partially) but not cyclins were suppressed in RT112 cells. AITC (but not BITC and PEITC) potently reduced Akt/pAkt and Raptor/pRaptor in drug-sensitive and drug-resistant T24 and RT112 cells. On the other hand, pRictor was elevated in drug-sensitive and drug-resistant T24 cells by AITC, whereas this mechanism was only true in the drug-resistant RT112 cells. Obviously, T24 are more sensitive to AITC treatment than RT112. This difference is also reflected by the cell growth assay (Figure 3), where the maximum tolerable AITC dosage was 70 µM in T24 but 140 µM in RT112 cell cultures. Attention should also be laid on the apoptosis assay. AITC caused strong elevation of apoptotic RT112 cells but a minor or no (gemcitabine-resistance) increase in T24 cell cultures. It is assumed that AITC exerts different modes of action on the tumor cells. Apoptosis induction may compensate minor effects on cell cycle protein alterations in RT112 cells, whereas down-regulation of cell cycle related pathways may compensate minor effects on apoptosis induction in T24 cells. This is still hypothetical; however, a similar scenario has recently been noted. AITC treatment reduced cell growth of both RT4 and T24 cells, albeit by a different mechanism. Cell death by apoptosis was the prominent underlying mechanism in RT4 cells, whereas blocking of cell cycle regulating proteins was the major action of AITC in T24 cells [17].

pAkt was reduced by all ITCs in the cisplatin- and gemcitabine-resistant tumor cells (not evaluated in BITC-treated T24 cells). This is notable, since genetic alterations within the PI3K/Akt/mTOR pathway have been noted in 42% of bladder cancer cases [28]. The PI3K-Akt pathway is closely involved in tumor development and is discussed to be a promising prognostic factor and therapeutic target in bladder cancer [29]. Recent analysis of tumor samples from TCGA databases revealed the PI3K/Akt pathway to be significantly related to metastatic bladder cancer [30]. The finding that pAkt was diminished in the resistant tumor cells by the ITCs is clinically relevant. Inactivating the PI3K/Akt pathway has just been shown to lower the risk of gemcitabine [31,32] and cisplatin resistance [33,34]. A clinical trial with the chemical Akt–mTOR inhibitor buparlisip has been started in this context, demonstrating modest activity in patients with platinum-resistant metastatic bladder cancer. Unfortunately, treatment with buparlisip was accompanied by severe toxicity [35].

The natural ITCs AITC, BITC, and PEITC distinctly reduced Akt phosphorylation in the in vitro model. In contrast to buparlisip or to ITC-related synthetic analogues, which were not qualified for clinical trials [36], they all have no (or low) toxicity in common. It might, therefore, be worthwhile to open clinical studies on bladder cancer patients integrating these plant-derived compounds into the GC-schedule.

The elevation of pRictor seen particularly in the drug-resistant RT112 and AITC-treated T24 cells requires final attention. Application of the HDAC inhibitor valproic acid to several prostate cancer cell lines down-regulated pRaptor but simultaneously up-regulated pRictor [37,38]. A recent investigation on hepatocellular carcinoma cells demonstrated that blockade of Akt enhanced phosphorylation of Rictor [39]. The relevance of this mechanism is not yet clear. It is to be emphasized Raptor and Rictor drive different cellular functions. Raptor is involved in the regulation of bladder cancer growth and proliferation, whereas Rictor serves as the main driving force of bladder cancer cell migration and invasion [40]. Since cell growth and proliferation of T24, RT112, and TCCSUP were all blocked by the ITCs, it seems not to be likely that enhancement of pRictor reflects a tumor escape phenomenon. However, it cannot be ruled out that activation of Rictor might be associated with an increased invasion behavior of the bladder cancer cells. It is mandatory to concentrate on this important issue in future experiments.

Concerning the cisplatin and gemcitabine non-responsiveness, we cannot finally assess to which extent the Akt–mTOR pathway and the cyclin–CDK axis might be involved in the process of resistance development. Akt–mTOR pathway activation has recently been shown to be associated with cisplatin-resistance in urothelial carcinoma cells, and suppression of it restored sensitivity to cisplatin in vitro and in vivo [34]. Akt–mTOR signaling has also been suggested to be involved in the acquisition of gemcitabin resistance [41]. Furthermore, network analysis of significantly altered proteins in bladder cancer revealed CDK2 as one central regulator mediating cisplatin resistance [42], with the consequence that targeting the CDK–cyclin axis may restore cisplatin-sensitivity [43]. It seems likely that the protein panel investigated here contributes to resistance acquisition. It should be explored next, whether BITC, AITC and/or PEITC may re-sensitize the bladder cancer cells to cisplatin and/or gemcitabine treatment.

## 4. Materials and Methods

### 4.1. Cell Culture

RT112, T24 (both: ATCC/LGC Promochem GmbH, Wesel, Germany) and TCCSupp (DSMZ, Braunschweig, Germany) bladder carcinoma cells were grown and cultured in Isocove’s Modified Dulbecco’s Medium (IMDM; Gibco/Invitrogen, Karlsruhe, Germany) supplemented with 10% fetal calf serum (FCS), 2% glutamax and 1% penicillin/streptomycin (all: Gibco/Invitrogen) in a humidified, 5% CO_2_ incubator. The cells were chosen do to their different grading. RT112 represents an invasive (pathological stage T2) moderately differentiated (grade 2/3) model of human bladder cancer. T24 is derived from a poorly differentiated (grade 3) bladder carcinoma, whereas TCCSupp were isolated from a grade 4 transitional cell carcinoma. Subcultures from passages 7–24 were selected for experimental use.

### 4.2. Resistance Induction

Cisplatin and gemcitabine (both provided by Hexal, Holzkirchen, Germany) were dissolved in DMSO at 1 mg/mL (cisplatin) or 20 µg/mL (gemcitabine) and stored at room temperature (cisplatin) or at 4 °C (gemcitabine). Resistance was induced by exposing parental cells to the drugs starting at 0.125 µg/mL (cisplatin) or 1.25 ng/mL (gemcitabine) and increasing stepwise to 1 µg/mL (cisplatin) or 10 ng/mL (gemcitabine-TCCSUP), or 20 ng/mL (gemcitabine-T24, RT112). This process took 3 to 6 months. To verify resistance, tumor cells were incubated for three days with cisplatin- or gemcitabine-free medium. Subsequently, different drug concentrations (0.125–4 µg/mL cisplatin or 1.25–40 ng/mL gemcitabine) were applied to the drug-resistant and drug-sensitive cells, and cell cultures were subjected to the MTT assay described below. Controls received cell culture medium alone. Cell lines were defined as resistant when their response to drug treatment was significantly reduced, compared to the response of the sensitive cell lines. Possible compound associated toxicity was determined by trypan blue (Gibco/Invitrogen). Once drug resistance had been achieved, all further investigation was carried out by comparing drug-sensitive to drug-resistant cells permanently exposed to 1 µg/mL cisplatin or 10 ng/mL gemcitabine (TCCSupp) or 20 ng/mL gemcitabine (RT112, T24).

### 4.3. Cell Growth

Cell growth was assessed using the 3-(4,5-dimethylthiazol-2-yl)-2,5-diphenyltetrazolium bromide (MTT) dye reduction assay (Roche Diagnostics, Penzberg, Germany). Bladder cancer cells (50 μL, 1 × 105 cells/mL) were treated with AITC, BITC or PEITC (all from Merck, Darmstadt, Germany) at different concentrations (versus non-treated) and then seeded onto 96-well tissue culture plates. After 24, 48, and 72 h, 10 μL MTT (0.5 mg/mL) were added for an additional 4 h. Cells were then lysed in a buffer containing 10% SDS in 0.01 M HCl. The plates were incubated overnight at 37 °C, 5% CO_2_. Absorbance at 550 nm was determined for each well using a microplate enzyme-linked immunosorbent assay (ELISA; Tecan Infinite M200, Männedorf, Switzerland) reader. Each experiment was performed in triplicate. After subtracting background absorbance, results were expressed as the mean cell number. The efficacy of treatment was then calculated based on control values set to 100%.

### 4.4. Apoptosis

To evaluate whether tumor cell growth was impaired or reduced due to apoptosis, the expression of annexin V/propidium iodide (PI) was evaluated using the annexin V-FITC Apoptosis Detection kit (BD Pharmingen, Heidelberg, Germany). In brief, tumor cells were washed twice with PBS, and then incubated with 5 μL of annexin V-FITC and 5 μL of PI in the dark for 15 min at room temperature. Cells were analyzed by flow cytometry using FACScalibur (BD Biosciences, Heidelberg, Germany). A total of 10,000 events were collected for each sample. The percentage of apoptotic (early and late), necrotic and vital cells in each quadrant was calculated using CellQuest software (BD Biosciences).

### 4.5. Clonogenic Growth

1000 single bladder cancer cells (treated with BITC (7.5 µM), AITC (20, 40 µM) PEITC (7.5 µM), versus non-treated) were transferred to 6-well plates. Following 5 to 10 days incubation without medium change, cell colonies were fixed and counted. Clones of at least 50 cells were counted as one colony.

### 4.6. Cell Cycle Analysis

Cell cycle analysis was carried out with sub confluent tumor cells after 24 and 72 h cultivation with or without BITC (7.5 µM), AITC (40 µM) PEITC (7.5 µM). Tumor cell populations were stained with propidium iodide, using a Cycle TEST PLUS DNA Reagent Kit (BD Biosciences, Heidelberg, Germany) and then subjected to flow cytometry with a FACScan flow cytometer (BD Biosciences). A total of 10,000 events were collected for each sample. Data acquisition was carried out using Cell-Quest software and cell cycle distribution was calculated using the ModFit software (BD Biosciences). The number of gated cells in the G1, G2/M, or S phases was presented as %.

### 4.7. Western Blot Analysis

To investigate the protein expression of cell cycle regulating proteins in T24 and RT112 cells (treated versus non-treated), tumor cell lysates were applied to a 7–12% polyacrylamide gel (depending on the protein size) and electrophoresed for 90 min at 100 V. The protein was then transferred to nitrocellulose membranes (1 h, 100 V). After blocking with nonfat dry milk for 1 h, the membranes were incubated overnight with monoclonal antibodies directed against the cell cycle proteins: CDK1/Cdc2 (IgG1, clone 1), pCDK1/Cdc2 (IgG1, clone 44/CDK1/Cdc2 (pY15)), CDK2 (IgG2a, clone 55), cyclin A (IgG1, clone 25), and cyclin B (IgG1, clone 18; all: BD Pharmingen). The mechanistic target of the rapamycin (mTOR) pathway was investigated using the following monoclonal antibodies: Raptor (clone 24C12), pRaptor (clone Ser 792), Rictor (clone D16H9), pRictor (clone Thr1135, clone D30A3; all: New England Biolabs), PKBα/Akt (IgG1 clone 55), phospho Akt (pAkt; IgG1, Ser472/Ser473, clone 104A282; both: BD Pharmingen). HRP-conjugated goat anti-mouse IgG and HRP-conjugated goat anti-rabbit IgG (both: 1:5000; Upstate Biotechnology, Lake Placid, NY, USA) served as the secondary antibody. The membranes were briefly incubated with ECL detection reagent (ECL; Amersham/GE Healthcare, München, Germany) to visualize the proteins and then analyzed by the Fusion FX7 system (Peqlab, Erlangen, Germany). β-Actin (1:1000; clone AC-15; Sigma-Aldrich, Taufenkirchen, Germany) served as the internal control. To quantify the intensity of the protein bands, the protein intensity/β-actin intensity ratio was calculated by the GIMP 2.8 software.

### 4.8. Statistics

All experiments were performed three to six times. Statistical significance was calculated with the Wilcoxon–Mann–Whitney-U test or the *t*-test. Differences were considered statistically significant at a *p*-value less than 0.05.

## 5. Conclusions

Evidence has been provided that ITCs may diminish the risk of cancer development, recurrence and progression. The in vitro data presented here point to the distinct tumor-suppressive properties of AITC, BITC, and PEITC in terms of growth and proliferation blockade. Thus, the therapeutic integration of an AITC/BITC/PEITC cocktail into the current bladder cancer treatment should be considered as a hopeful strategy to enhance patient outcome. The results point to a different molecular action of the compounds. This may offer an advantage to synthetic drugs, since targeting several molecules may prevent rapid therapeutic resistance. Further investigations are however necessary to explore the full potential of the ITCs, with particular emphasis on resistance development.

## Figures and Tables

**Figure 1 ijms-23-10996-f001:**
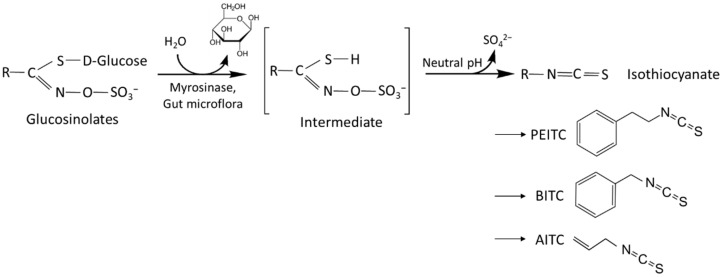
Schedule of PEITC, BITC and AITC generation by enzymatic hydrolysis.

**Figure 2 ijms-23-10996-f002:**
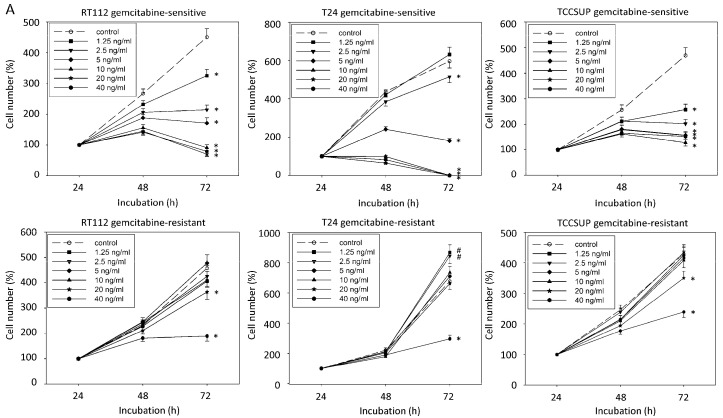
Growth blocking effects of increasing concentrations of gemcitabine (up: sensitive cells, down: resistant cells), (**A**), and cisplatin (up: sensitive cells, down: resistant cells), (**B**). Cell number was evaluated after 24, 48, and 72 h by the MTT assay, whereby 24 h values were set to 100%. Error bars indicate standard deviation. Experiments were repeated five times. One representative experiment is shown. * indicates significant down-regulation, # indicates significant up-regulation to the untreated control, *n* = 6.

**Figure 3 ijms-23-10996-f003:**
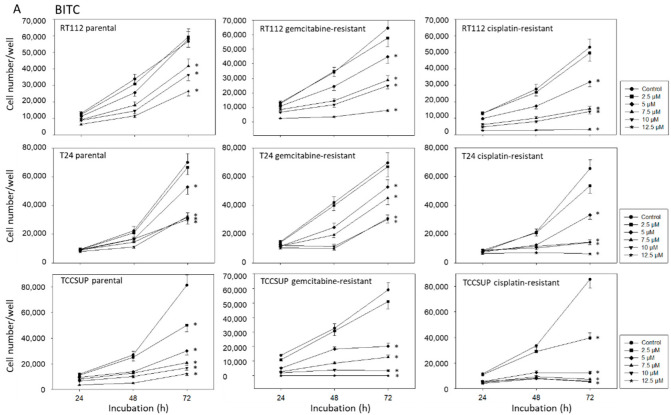
(**A**–**C**) Influence of BITC (**A**), AITC (**B**) and PEITC (**C**) on growth of parental, cisplatin- and gemcitabine-resistant RT112, T24, and TCCSUP bladder cancer cell lines. Cell number was evaluated after 24, 48, and 72 h by the MTT assay. Error bars indicate standard deviation (SD), *n* = 5. * indicates a significant difference to the non-treated control.

**Figure 4 ijms-23-10996-f004:**
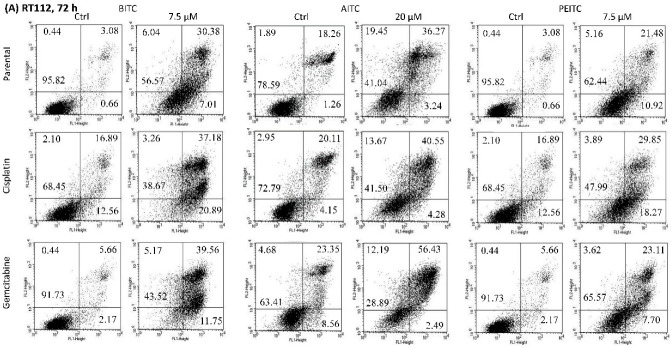
Early apoptosis, late apoptosis and necrosis of parental and resistant RT112 (**A**) and T24 cells (**B**) treated with BITC, AITC or PEITC for 72 h. Controls (Ctrl) remained untreated. Upper left quadrants show the percentage of cells in necrosis, upper right quadrants show the percentage of cells in late apoptosis, lower right quadrants show the percentage of cells in early apoptosis and lower left quadrants vital cells (one representative of 3 analyses; SD_intra-assay_ < 10%). A total of 10,000 events were collected for each sample.

**Figure 5 ijms-23-10996-f005:**
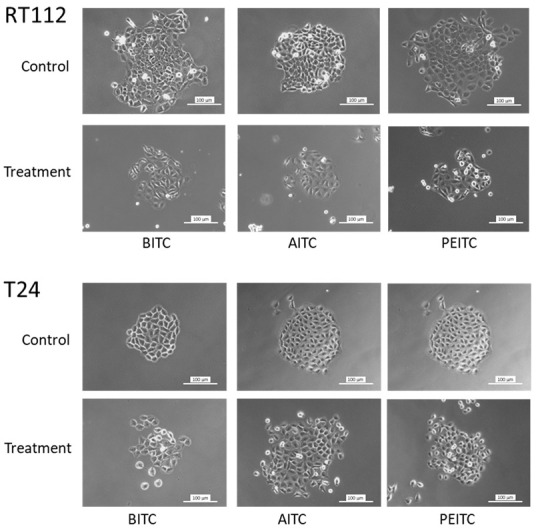
Influence of BITC (7.5 µM), AITC (20 µM) and PEITC (7.5 µM) on clone formation of parental RT112 and T24 cells. Controls were without drugs. Scale bar: 100 µm.

**Figure 6 ijms-23-10996-f006:**
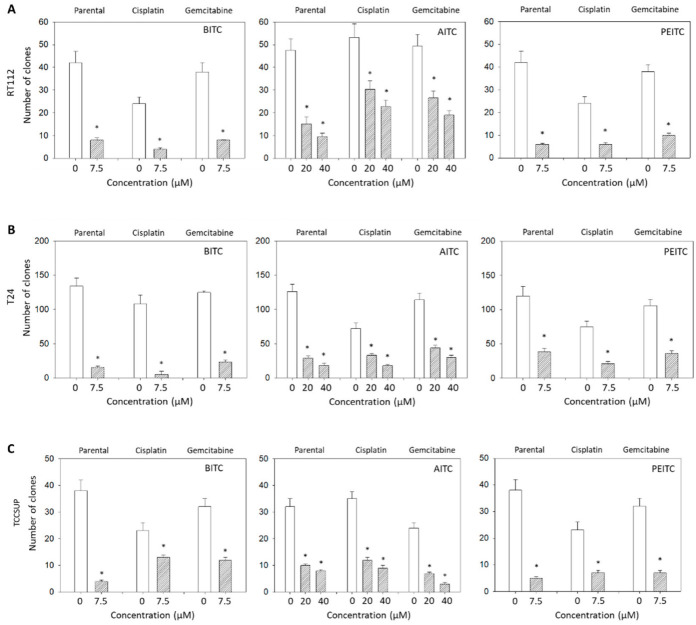
Number of RT112 (**A**), T24 (**B**), and TCCSUP (**C**) clones (parental, cisplatin-resistant, gemcitabine-resistant) exposed to BITC (7.5 µM), AITC (20, 40 µM) or PEITC (7.5 µM), indicated as stripped bars. Controls remained untreated (white bars), *n* = 3. * indicates a significant difference to untreated controls.

**Figure 7 ijms-23-10996-f007:**
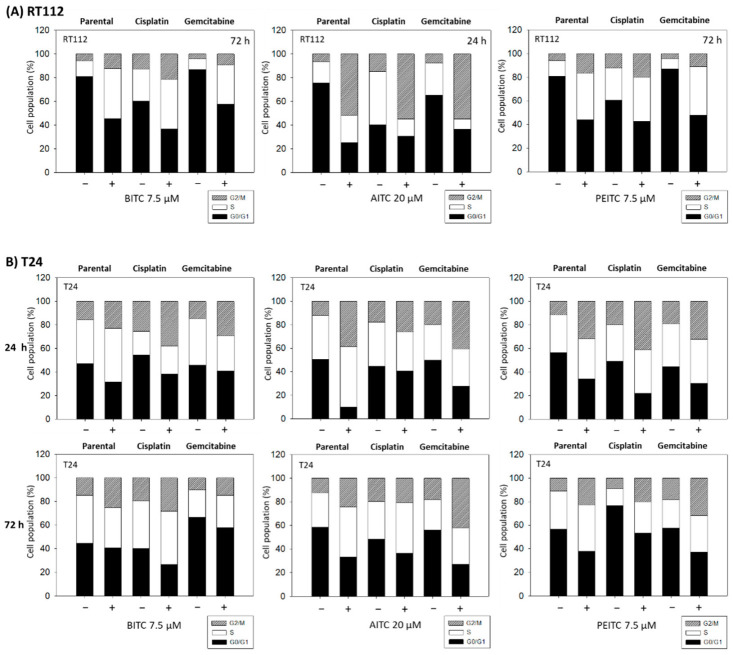
Cell cycle distribution in parental, cisplatin- or gemcitabine-resistant RT112 (**A**) and T24 cells (**B**) following BITC (7.5 µM), AITC (20 µM) or PEITC (7.5 µM) exposure for 24 and/or 72 h. Controls (−) remained untreated. “+” indicates drug treatment. One representative of three separate experiments is shown.

**Figure 8 ijms-23-10996-f008:**
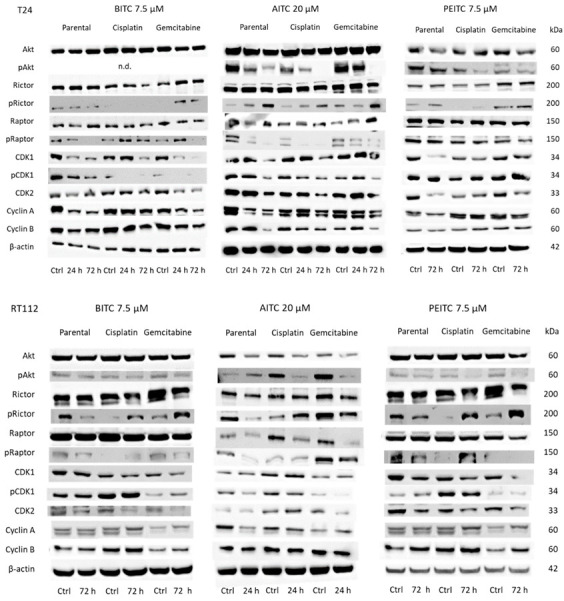
Western blot of cell cycle- and mTOR-related proteins from RT112 and T24 lysates of parental and cisplatin/gemcitabine-resistant cells. Tumor cells were pretreated with BITC (7.5 µM), AITC (20 µM) or PEITC (7.5 µM) for 24 and/or 72 h. Controls (Ctrl) remained untreated. Β-actin served as the internal control. One representative from three separate experiments.

**Figure 9 ijms-23-10996-f009:**
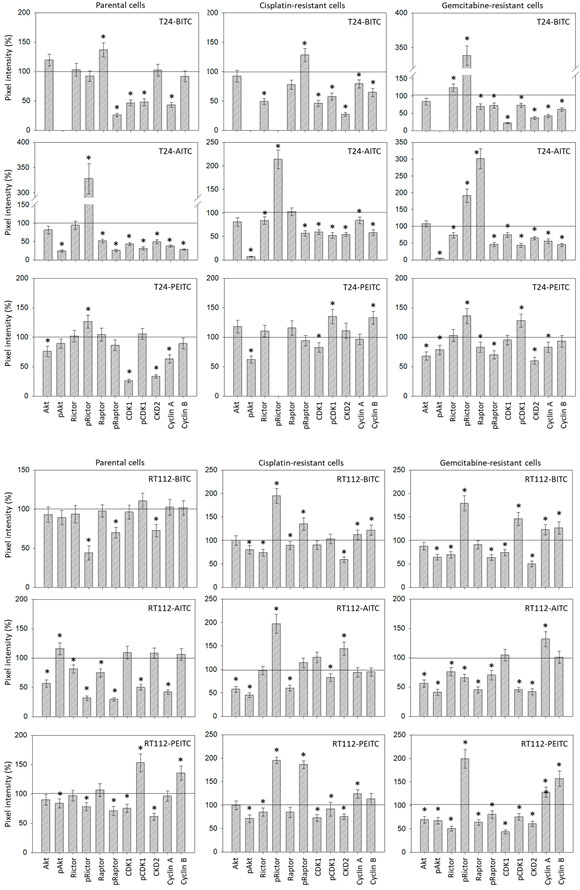
Pixel density analysis of the protein level in the sensitive and resistant bladder cancer cell lines cells following SFN treatment. Values are given in percentage, related to the 100% control (indicated by a black line). * significant difference to controls.

## Data Availability

Not applicable.

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
