# Peer review of "Allyl-, Butyl- and Phenylethyl-Isothiocyanate Modulate Akt–mTOR and Cyclin–CDK Signaling in Gemcitabine- and Cisplatin-Resistant Bladder Cancer Cell Lines"

_ijms, 2022, doi:10.3390/ijms231910996_

Round 1

Reviewer 1 Report

  Rutz et al. investigated the functions of some natural chemicals on the gemcitabine- and cisplatin-resistant bladder cancer cells lines. They used FACS and expression analyses by western blotting and found mTOR and cyclin-CDK signalling pathways are influenced.

The observations are overall intriguing and the manuscript is well prepared. I would suggest a few points that can improve the readability of the manuscript.

(1) line 175: subheading: Influence of… is too abstract. What is the main claim of this part? Please summarise the results in the subheading. This is the case for line 195.

(2) There are many findings in each figure, and it is difficult to follow which figure corresponds to the text statement. I would suggest to label A,B,C,… on each graph and FACS data, and indicate the correspondence clearly. Also, western blot data (Fig8) are difficult to follow. Please highlight the altered expression with rectangles.

Author Response

Comment 1: 1) line 175: subheading: Influence of… is too abstract. What is the main claim of this part? Please summarise the results in the subheading. This is the case for line 195. Our answer: We changed the subheadings as suggested. Heading of chapter 2.5 reads now: “Cell cycle arrest by BITC, AITC, and PEITC”. Heading of chapter 2.6 reads: “Alteration of cell cycle protein expression”. Comment 2: There are many findings in each figure, and it is difficult to follow which figure corresponds to the text statement. I would suggest to label A,B,C,… on each graph and FACS data, and indicate the correspondence clearly. Also, western blot data (Fig8) are difficult to follow. Please highlight the altered expression with rectangles. Our answer: Figure 4 has now been divided into 4A and 4B. The respective manuscript text also refers now to 4A and 4B (“Data are depicted representatively for RT112 (figure 4A) and T24 cells (figure 4B), exposed to the compounds for 72 h”). The same has been done with figure 6 and 7. In fact, hugh amount of data are shown in figure 8. To allow a better overview and interpretation, data have been quantified by a pixel density analysis and presented in figure 9.

Reviewer 2 Report

This study investigated the effect of isothiocyanates (ITCs), allyl-isothiocyanate (AITC), butyl-isothiocyanate (BITC), and phenylethyl-isothiocyanate (PEITC) on gemcitabine- and cisplatin-resistant bladder cancer cells lines. There are several comments to the authors.

1.     Why these drugs were applied in bladder cancers should be mentioned.

2.     mg/ml or mM should be selected to avoid confusing the readers.

3.     The quantified results of Figure 4 should be provided.

4.     Two concentration is suggested in Figure 4 and 5.

5.     The quality of figures should be improved. Several lines were observed in the figures.

6.     The quality of western blots should be improved. Drug treatment time should be the same. Quantification should be provided.

7.     Why were these three cell lines selected should be described.

8.     All the results looked like preliminary data. The authors should improve the results and present better.

9.     These drugs improved apoptosis and arrested cell cycle. However, did these drugs suppress the reason that caused bladder cancer cells to be gemcitabine- or cisplatin-resistant cells? The mechanism should be clarified.

Author Response

Comment 1: Why these drugs were applied in bladder cancers should be mentioned. Our answer: We have now included more information about the role of isothiocyanates in bladder cancer. Introduction, last para reads now “Evidence has already been provided that AITC and BITC suppress bladder cancer growth and enhance apoptosis in vitro. Both compounds reduced the size of bladder tumors and prevented oncogenesis in a rat model as well. PEITC is also assumed to play critical roles in preventing the initiation step of carcinogenesis and inhibiting tumor progression. However, the detailed molecular mechanisms have not been clearly defined [12]. There are also no publications available dealing with the influence of the ITCs on cisplatin- and gemcitabine-resistant bladder cancer”. Comment 2: mg/ml or mM should be selected to avoid confusing the readers. Our answer: We would like to note that concentrations of gemcitabine and cisplatin were given in ng/ml or µg/ml (fig. 2), whereas AITC, BITC and PEITC concentrations are related to µM throughout the manuscript. Figure 2 just served to demonstrate cisplatin- and gemcitabine-resistance, and concentration has not been mentioned thereafter. Still, to prevent confusions, ng and µg have now been correlated to µM: 1 µg/ml cisplatin = 3,332 µM; 20 ng/ml gemcitabine = 0,0758 µM (please see “Results”, chapter “2.1. Resistance induction”). Comment 3: The quantified results of Figure 4 should be provided. Our answer: Each event has been related to 10.000 cells. We apologize that this information has not been included in the methods part. The chapter “apoptosis” reads now: “Cells were analyzed by flow cytometry using FACScalibur (BD Biosciences, Heidelberg, Germany). 10,000 events were collected for each sample. The percentage of apoptotic (early and late), necrotic and vital cells in each quadrant was calculated using CellQuest software (BD Biosciences)”. This information is also given in the respective figure legend (figure 4). Comment 4: Two concentration is suggested in Figure 4 and 5. Our answer: This is correct. In fact, analysis of the ITCs demonstrated different efficacy on tumor growth (please see chapter “2.2. BITC, AITC, and PEITC block growth of drug-resistant tumor cells”). Based on the results shown in figure 3, we defined the working concentration to be 7.5 µM for BITC and PEITC, and 20 µM for AITC. In this context, the last sentence of this chapter reads “All further experiments were then done with 7.5 µM BITC, 20 µM AITC (if not otherwise indicated), and 7.5 µM PEITC”. Comment 5: The quality of figures should be improved. Several lines were observed in the figures. Drug treatment time in Western blots should be the same. Quantification should be provided. Our answer: We apologize for the bad quality of the pre-print version of the manuscript. This should be improved in the final version (as it was told to us by the MDPI office). PEITC did not alter cell cvcle proteins in T24 cells after 24h (as controlled by Akt, CDK1, CDK2, Cyclin A+B). We, therefore, did not carry out detailed experiments after this time point but rather concentrated on the 72 h incubation period. The same was done with the analysis of PEITC caused alterations in RT112 cells. In contrast, AITC induced potent alterations already after 24 h in RT112 cells. This observation has been taken care of in the results section which reads: “Concerning RT112, maximum effects were seen after 72 h in presence of BITC and PEITC, whereas strongest modulation was evoked by AITC after 24 h”. We, therefore, would not like to include further Western blot data demonstrating no differences not to further complicate the figure. In this context, there was also concern of referee 1 who noted that “western blot data (Fig8) are difficult to follow”. Quantification by pixel analysis has been done and included (please see figure 9). Comment 6: Why were these three cell lines selected should be described. Our answer: The cell lines we used are characterized by a different grading which allowed us to evaluate whether the efficacy of the ITCs may depend on the differentiation status of the tumor cells. Based on our data, this was obviously not the case. The chapter “Cell culture” reads now: “The cells were chosen do to their different grading”. Comment 7: All the results looked like preliminary data. The authors should improve the results and present better. These drugs improved apoptosis and arrested cell cycle. However, did these drugs suppress the reason that caused bladder cancer cells to be gemcitabine- or cisplatin-resistant cells? The mechanism should be clarified. Our answer: We do not agree that our data are preliminary. Indeed, several experiments have been carried out to demonstrate resistance induction, suppression of cell growth and proliferation, apoptosis, and cell cycle arrest triggered by three different ITCs, and to demonstrate alterations of cell cycle relevant pathways. Please not that three cell lines have been used, each drug-sensitive, cisplatin- and gemcitabine-resistant. A huge amount of data sets are presented in 9 figures. Concerning drug resistance, we have already pointed to Akt as a potential driver of cisplatin- and gemcitabine-resistance (please see reference 32-35). Still, resistance mechanism are quite complex. Aside from influx-efflux-alterations, epigenetic changes, disturbances of protein trafficking and defective cytoskeleton organization, miRNA expression profile and activation/deactivation of several intracellular pathways may all be associated with resistance induction. We believe that it might be far beyond the scope of the present article to analyze all the mechanism involved in resistance development (which would also involve respective experiments with ITC treated cells). However, we have included a further paragraph in the discussion section dealing with the Akt-mTOR pathway and the CDK-Cyclin-axis in the context of cisplatin-/gemcitabine-resistance. The final paragraph reads: “Concerning the cisplatin and gemcitabine non-responsiveness, we cannot finally assess to which extent the Akt-mTOR pathway and the Cyclin-CDK-axis might be involved in the process of resistance development. Akt-mTOR pathway activation has recently been shown to be associated with cisplatin-resistance in urothelial carcinoma cells, and suppression of it restored sensitivity to cisplatin in vitro and in vivo [42]. Akt-mTOR signaling has also been suggested to be involved in the acquisition of gemcitabin resistance [43]. Furthermore, network analysis of significantly altered proteins in bladder cancer revealed CDK2 as one central regulator mediating cisplatin resistance [44], with the consequence that targeting the CDK-Cyclin-axis may restore cisplatin-sensitivity [45]. It seems likely that the protein panel investigated here contributes to resistance acquisition. It should be explored next, whether BITC, AITC and/or PEITC may re-sensitize the bladder cancer cells to cisplatin and/or gemcitabine treatment”. References have been added in the reference list.

Round 2

Reviewer 2 Report

1. The square lines of each subfigure should be delete.

2. Scale bars are needed in Figure 5.

Author Response

Comment 1: The square lines of each subfigure should be delete. Our answer: This comment is not clear to us. Is it related to figure 9? A line has been included here to better demonstrate the 100% control value. We are prepared to remove this line if this has been addressed by the referee. Comment 2: Scale bars are needed in Figure 5. Our answer: Scale bars were included. Figure legend reads: “Scale bar: 100 µm”.